# Medusa: Software to build and analyze ensembles of genome-scale metabolic network reconstructions

**Gregory L. Medlock** [1¤]*, **Thomas J. Moutinho** [1], **Jason A. Papin** [1,2,3]*

**1** Department of Biomedical Engineering, University of Virginia, Charlottesville, Virginia, United States of America, **2** Department of Medicine, Division of Infectious Diseases and International Health, University of Virginia, Charlottesville, VA, United States of America, **3** Department of Biochemistry and Molecular Genetics, University of Virginia, Charlottesville, Virginia, United States of America

¤ Current address: Division of Gastroenterology, Hepatology, and Nutrition, Department of Pediatrics, University of Virginia, Charlottesville, Virginia, United States of America
* glm5uh@virginia.edu (GLM); papin@virginia.edu (JAP)

**Data Availability Statement:** The data underlying the results presented in the study are available from https://github.com/opencobra/medusa.

**Funding:** We acknowledge funding from the National Institutes of Health R01GM108501,

## Abstract

Uncertainty in the structure and parameters of networks is ubiquitous across computational biology. In constraint-based reconstruction and analysis of metabolic networks, this uncertainty is present both during the reconstruction of networks and in simulations performed with them. Here, we present Medusa, a Python package for the generation and analysis of ensembles of genome-scale metabolic network reconstructions. Medusa builds on the COBRApy package for constraint-based reconstruction and analysis by compressing a set of models into a compact ensemble object, providing functions for the generation of ensembles using experimental data, and extending constraint-based analyses to ensemble scale. We demonstrate how Medusa can be used to generate ensembles and perform ensemble simulations, and how machine learning can be used in conjunction with Medusa to guide the curation of genome-scale metabolic network reconstructions. Medusa is available under the permissive MIT license from the Python Packaging Index (https://pypi.org) and from github (https://github.com/opencobra/Medusa), and comprehensive documentation is available at https://medusa.readthedocs.io/en/latest.

This is a *PLOS Computational Biology* Software paper.

## Introduction

Hypothesis-driven computational models of biological systems are being increasingly applied to guide experimentation [1]. In hypothesis-driven modeling, in contrast to data-driven modeling [2], hypothesized biological parts, functions, and interactions are mathematically formalized to allow *in silico* experimentation. These models take many forms, ranging in complexity from a single linear equation relating two quantities to systems of nonlinear differential equations describing dynamic systems.

R01AT010253, and T32LM012416 to J.P. We also acknowledge funding from the Thomas F. and Kate Miller Jeffress Memorial Trust to J.P., a Wagner predoctoral fellowship to G.L.M., and a Grand Challenges Exploration Phase I grant (OPP1211869) from the Bill & Melinda Gates Foundation to G.L.M. The funders had no role in study design, data collection and analysis, decision to publish, or preparation of the manuscript.

**Competing interests:** The authors have declared that no competing interests exist.

Across all hypothesis-driven modeling frameworks, the choice of model scope and parameter values may strongly influence simulation results. For some types of hypothesis-driven models in biology, approaches from other fields have been applied to quantify the influence of parameter values on simulation outcomes, such as sensitivity analysis of dynamical models [3]. For network-based models of biological systems such as metabolic or signaling networks, the presence or absence of a network component may be uncertain due to lack of characterization or uncertainty in data itself. Traditional sensitivity analysis methods have recently been reformulated for these systems to analyze sensitivity to topological variation, but these methods have not seen wide adoption [4]. While uncertainty in network structure poses analytical difficulties, it also presents an actionable framework to accelerate biological discovery. Alternative network structures can guide experimental design, allowing comparison of simulation results for alternative networks to experimental data to identify the network structure most consistent with biological behavior (i.e. model selection) [5]. This uncertainty can also be used to prioritize experiments that will maximally improve confidence in the simulations performed with a model (i.e. uncertainty reduction) [6].

In studies of metabolism, genome-scale metabolic network reconstructions (GENREs) have emerged as a useful formalism for hypothesis-driven modeling [7]. In conjunction with biological objective functions, such as maximization of growth rate, GENREs can be used to construct genome-scale metabolic models (GEMs). In addition to topological uncertainty (e.g., presence/absence of reactions in a network), simulations with GEMs generally yield many alternative solutions. Even the simplest simulations that can be performed with GEMs exhibit this behavior. This is the case for flux balance analysis (FBA), in which a pseudo-steady state is assumed, and flux values are found for all reactions in a GEM such that an objective function is optimized [8]. While a single global maximum value for the objective is guaranteed to be found, flux through every other component of the network is only constrained within a solution space, not to a single value. As a result, even though performing FBA yields a single value for the flux through reactions in a network, there are an infinite number of feasible flux values within the range determined by the solution space for some reactions. Techniques such as flux variability analysis and flux sampling have been developed to explore the space of alternative solutions in this scenario [9,10].

A myriad of additional algorithms have been developed for the analysis of GEMs for strain engineering, contextualization of experimental data, and building cell- and tissue-type specific GEMs [11–13]. In addition to optimization problems that can be solved using linear programming such as FBA, problems have been formulated to take advantage of mixed integer linear programming (MILP; see [14] for a review of optimization problems in systems biology). MILP employs binary state variables during optimization to solve problems that involve discrete activation or inactivation of variables. MILP problems are particularly well-suited to network-based models, since they allow switch-like behavior that can include or exclude network components (e.g., shutting reactions off/on). MILP has been used widely for gap-filling of GEMs, a process in which constraints or objectives are set to recapitulate a known phenotype by adding biochemical functions from a universal set of reactions [15]. In MILP problems used for gap-filling, the objective function is generally minimization of the number of modifications to a GEM that must be made to satisfy the constraints imposed (e.g., metabolite uptake or secretion, production of biomass). One consequence of this formulation is that alternative solutions, which contain unique sets of reactions which need to be altered in the network or added, are common for large networks that have a large space of potential solutions to draw from (e.g., a large universal set of reactions). These alternative optima in MILP problems are increasingly being considered and leveraged to understand redundancy in solutions and whether or not portions of a solution may be spurious [16–18].

It has been shown that the order in which separate instances of gap-filling are applied to the same network (e.g., gap-filling for growth on individual carbon sources iteratively) strongly influences which reactions are included in the resulting network [19]. In this same study, the alternative solutions generated during this process were used to improve gene essentiality predictions using EnsembleFBA, a technique in which sets of alternative GEMs are used to perform FBA to determine gene essentiality. Using the entire ensemble, performance can be tuned by varying the voting threshold required to make a specific prediction. This is analogous to the threshold-based voting procedure used to construct receiver operating characteristic curves for ensemble-based machine learning models such as random forest [20]. This approach is likely to be highly beneficial for studies of organisms for which little biochemical data are available, which typically have many gaps in their GENRE and thus have many highly-variable alternative gap-filling solutions. Although a nascent approach for studying GENREs, we have built on these observations, and ensemble generation and analysis have been applied in several cases [6,19,21,22].

Here, we present Medusa, a Python package for the generation and analysis of ensembles of GENREs. Medusa provides a framework for compactly representing ensembles of GENREs, avoiding the redundancy of storing many separate models while still being flexible enough to represent variation in any component within a GENRE. Medusa manages ensemble storage and indexing during simulation, allowing users to interact with an entire ensemble in the same way they would interact with an individual GENRE using any constraint-based reconstruction and analysis (COBRA) method. Furthermore, by standardizing the representation of ensembles and their interface with existing COBRA methods, Medusa enables the application of supervised and unsupervised machine learning to gain insight into the influence of varying components within an ensemble of GENREs on the predictions they make. The architecture and functionality of Medusa were designed to make ensemble analyses as accessible and usable as COBRA methods applied to single networks.

## Design and implementation

### Architecture overview and dependencies

Medusa is built on top of COBRApy, a Python-based package in which many COBRA methods are implemented [23]. Although a dependency-free approach in which ensemble simulation methods are implemented from the ground up could be more efficient, we chose to extend COBRApy to greatly decrease the size and complexity of the codebase and to reduce the domain-specific knowledge required to use Medusa and understand the source code (i.e. decrease the effort for existing COBRApy users and contributors to use Medusa). As such, the architecture of Medusa closely mimics COBRApy **Fig 1**.

At the time of this writing, GENREs are represented within COBRApy using a Model class. The Model class manages the interface between COBRApy and numerical solvers through optlang, a Python package for formulating and solving optimization problems that extends the symbolic mathematics package SymPy [24,25]. GENREs are represented by a Model using additional classes with biological analogs (Metabolite, Reaction, and Gene). Objects belonging to each of these classes are stored within container-like objects (metabolites, reactions, and genes, respectively) that are each an attribute of a Model. Each Metabolite, Reaction, and Gene has attributes which might affect simulations performed using the Model, such as the lower and upper bounds of flux through each Reaction or the gene-protein-reaction relationship for each Reaction, which link them to specific Genes.

In Medusa, ensemble functionality is introduced using three new classes. The first, Feature, describes a GENRE component which has a parameter that varies across an ensemble (e.g., a

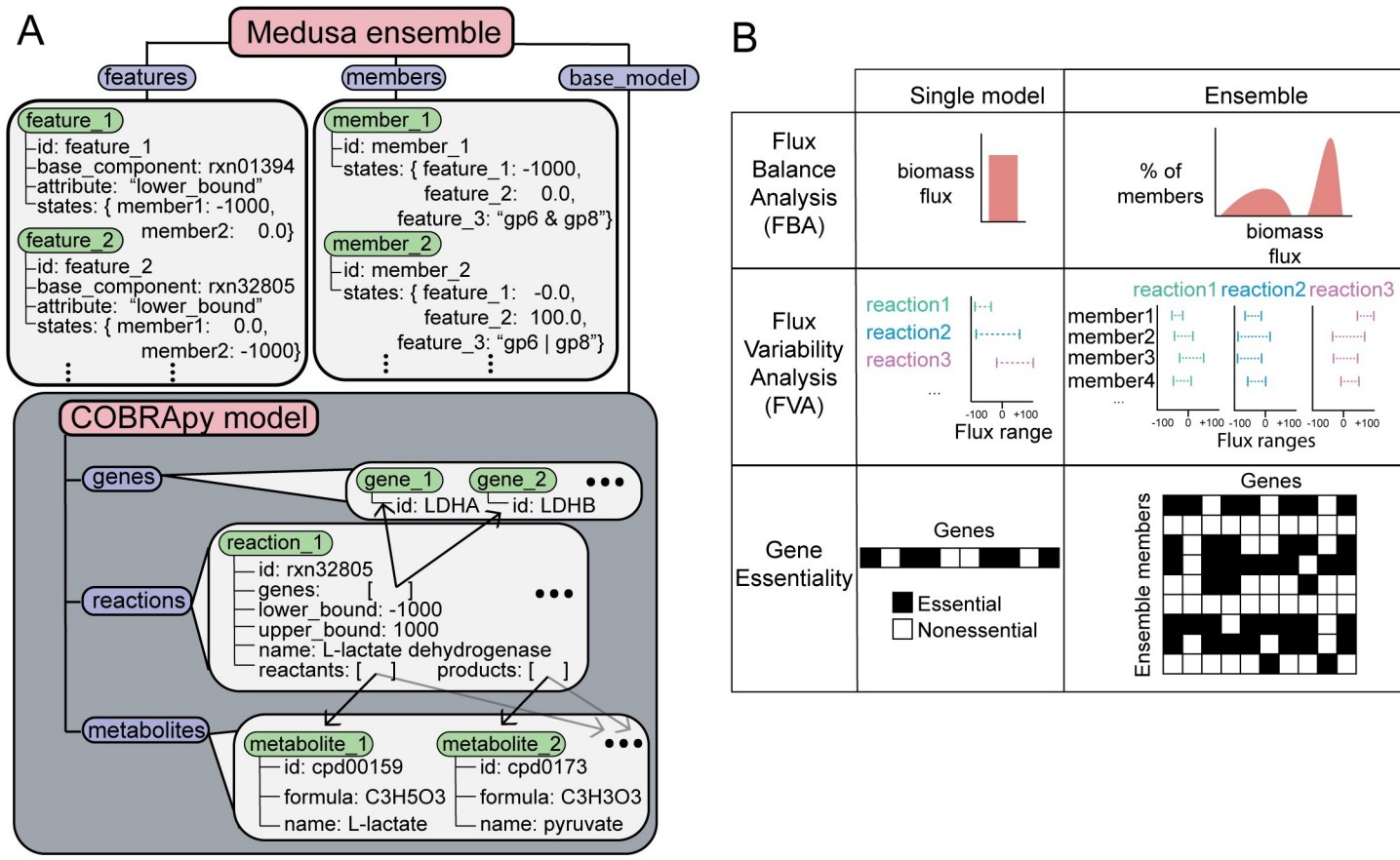

**Fig 1. Architecture of ensembles and simulation output in Medusa. A)** Ensemble functionality is implemented in Medusa through the Ensemble class. The Ensemble class exposes three attributes to the user: features, members, and base_model. Both features and members are container-like objects similar to genes, reactions, and metabolites in COBRApy. Within each container, Medusa objects of class feature and member are stored. The base_model attribute points to a COBRApy model. This base_model contains all of the features present in any member of the ensemble, and is manipulated when generating an ensemble or performing simulations. **B)** Description and shape of simulation results for common COBRA methods as implemented in Medusa. Each simulation method returns a distribution-equivalent to the single-model simulation result.

reaction that is reversible in some ensemble members but irreversible in others). The second, Member, describes individual GENREs within an ensemble and their state for each Feature. The third is the Ensemble class, which references every Feature and Member associated with an ensemble of GENREs, as well as a COBRApy Model, referred to as the "base model". This base model holds all COBRApy objects that might be associated with any Member in the Ensemble. Within an Ensemble, each Feature references the component within the base model (Metabolite, Reaction, or Gene) for which it encodes alternative parameter values, as well as the attribute within that component that is modified (e.g., the upper bound of flux through a reaction). When a simulation is to be performed using a particular GENRE within an Ensemble, Medusa changes the state of the base model to represent the proper state for the corresponding Member for every Feature. Thus, the Ensemble can represent any number of variants in GENRE structure throughout an ensemble (e.g., reaction presence/absence, reversibility, alternative gene-protein-reaction relationships) and can be used to apply any methods implemented in COBRApy. Furthermore, this implementation has a memory footprint only slightly larger than a single COBRApy Model, and facilitates queuing of simulations for parallel processing.

Medusa is developed partially with test-driven development. Unit tests are implemented using the pytest package (https://docs.pytest.org/en/latest) and are run automatically with each modification to the Medusa github repository via continuous integration with TravisCI (Travis CI, GMBH, Berlin, Germany). Support is provided for Python version 3.4 and later. Tests are run in TravisCI using Python 3.6, with plans to include Python 3.7 and 3.8. No support is provided for Python 2.

## Results

### Performing ensemble simulations

Currently, users can perform FBA, flux variability analysis (FVA), single gene deletions, and single reaction deletions using an ensemble in Medusa. In each case, the simulations are performed with a single function, which returns a collection of results, where each entry corresponds to the simulation results for a single ensemble member. Users have the option of performing simulations using the entire ensemble, a specific set of Members, or a random fraction of Members. For the currently implemented analyses, the collection of results are returned as a DataFrame from the pandas Python package, where each column corresponds to the entry normally populating the results for a single network (e.g., a reaction ID for FBA/FVA, a gene ID for single gene deletions), and each row corresponds to an ensemble Member (except for FVA, where two columns are required for each ensemble member to describe the minimum and maximum fluxes). See **Fig 2** for a schematic describing how the shape of data describing simulation results changes for each simulation method.

Because the Ensemble object implemented in Medusa maintains a COBRApy Model object, users can also perform any custom simulation they would like by 1) manipulating the COBRApy Model to be suitable for their simulations (e.g., set custom constraints or objectives), 2) setting the state of the model to represent an ensemble member using Medusa functionality, 3) performing their simulation, then 4) iterating through any other ensemble members they would like and performing steps 2–3.

### Comparing ensemble simulations

When performing simulations with an ensemble rather than a single model, results shift from single values to distributions. While this explicitly accounts for uncertainty, it also requires that statistical approaches are applied to interpret differences in distributions. In the simplest case, performing ensemble FBA to predict the growth rate on each of two different media conditions for a single bacterial species generates two distributions of predicted growth rates **Fig 3A**. A paired univariate test (e.g., t-test or a non-parametric equivalent) can be used to determine whether the predicted growth rate is equal in these two conditions. This example is demonstrated in the Medusa documentation at https://medusa.readthedocs.io/en/latest/stats_compare.html.

### Coupling ensemble modeling with machine learning

The availability of ensemble generation and simulation methods in Medusa provides ample opportunity to apply machine learning to leverage variation in ensembles. One application area that we have developed focuses on guiding the curation of genome-scale metabolic network reconstructions by attributing simulation uncertainty to network components in the ensemble, then prioritizing curation of these components based on how much they contribute to simulation uncertainty [6]. This approach can be broken down into four steps: 1) ensemble generation, 2) ensemble simulations, 3) unsupervised learning to summarize simulation

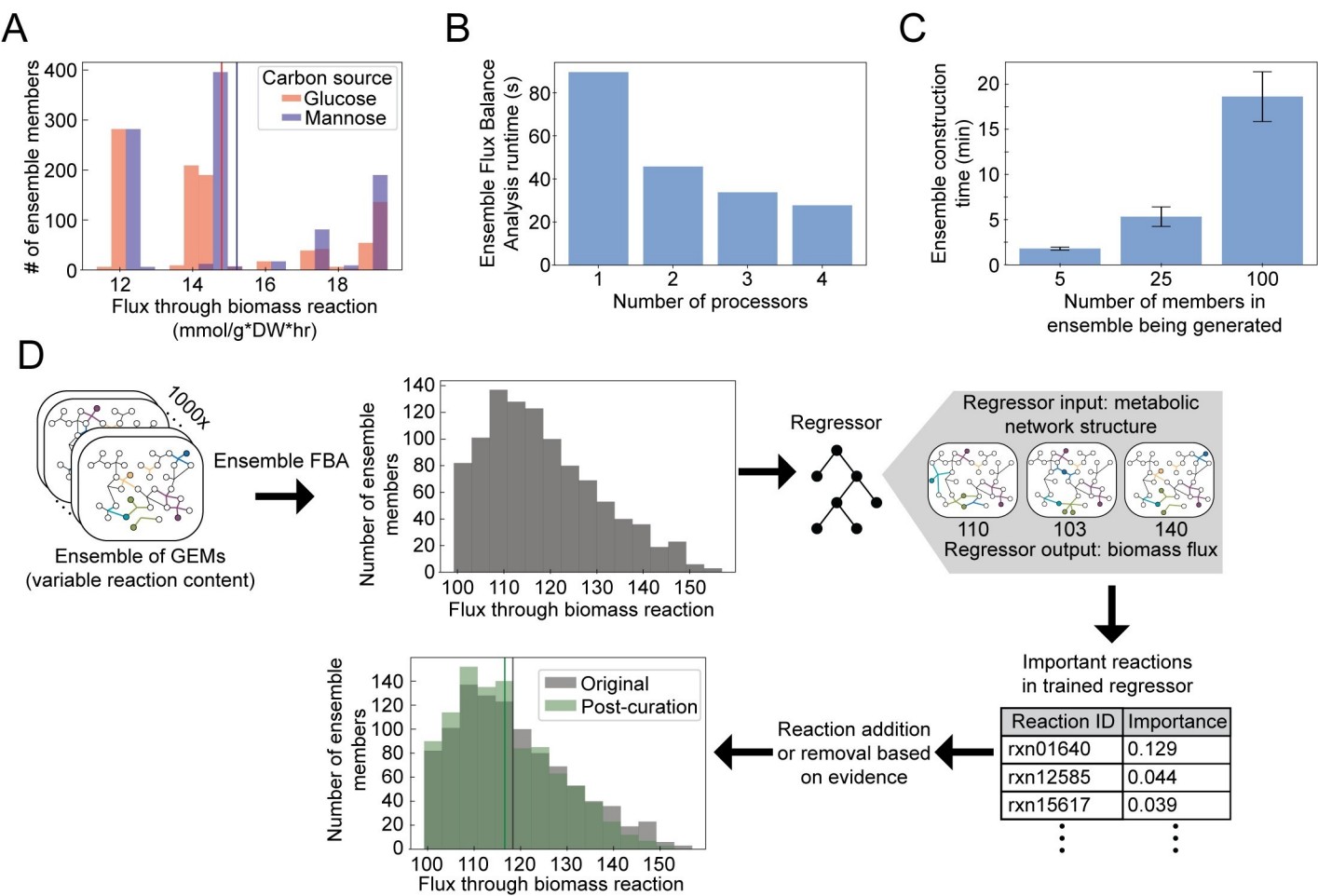

**Fig 2. Usage and benchmarking of Medusa. A) Comparison of ensemble flux balance analysis (FBA) simulations across different conditions**. **A)** Ensemble FBA performed on glucose or mannose minimal media using an ensemble of 1000 GEMs for *Staphylococcus aureus*. This ensemble was generated in [6] by iteratively gap-filling a draft reconstruction to enable biomass production in single-carbon source growth conditions supported with experimental data. Mean for the distribution for either condition shown by vertical line of same color. Predicted flux through biomass is higher on mannose than glucose (Wilcoxon signed-rank test, $p <$ 1E-5). **B)** Benchmarking of Ensemble FBA. Simulations were performed with the same *S. aureus* ensemble of 1000 members as in panel A. **C)** Benchmarking of ensemble generation time with iterative gapfilling (process shown in Fig 3). **D** Comparison of Ensemble FBA simulations before and after curation using a machine learning-guided approach. Ensemble FBA was performed on complete medium (uptake of -1000 mmol/g*DW*hr allowed for all metabolites) using an ensemble of 1000 GEMs for *S. aureus*. Mean for the distributions before and after curation shown by vertical line of same color in the last panel in the workflow. Ensemble and machine-learning guided curation identified N-Formimino-L-glutamate iminohydrolase as a driver of variation in simulated flux through biomass. Based on a literature search, this reaction was inactivated in all ensemble members and ensemble FBA was performed again, resulting in the shift in the distribution shown. See documentation for full narrative-style example: https://medusa.readthedocs.io/en/latest/machine_learning.html.

uncertainty, and 4) supervised learning to associate uncertainty in network structure with uncertainty in simulations. In addition to our published work utilizing Medusa for this purpose, we provide an example in the Medusa documentation that applies this method to a single ensemble FBA simulation: https://medusa.readthedocs.io/en/latest/machine_learning.html.

In this example, a previously generated ensemble is loaded into Medusa, media conditions are set to allow uptake of any metabolites with transporters in the ensemble, and ensemble FBA is performed. Then, a random forest regressor is used to predict the simulated values for flux through biomass (e.g., the values generated with ensemble FBA) for each ensemble member using the Medusa states (i.e., binary reaction presence/absence) for the same ensemble member as input. Examining the most predictive feature in the random forest regressor, we perform a literature search and find that the feature is likely not present for the bacterial species under study

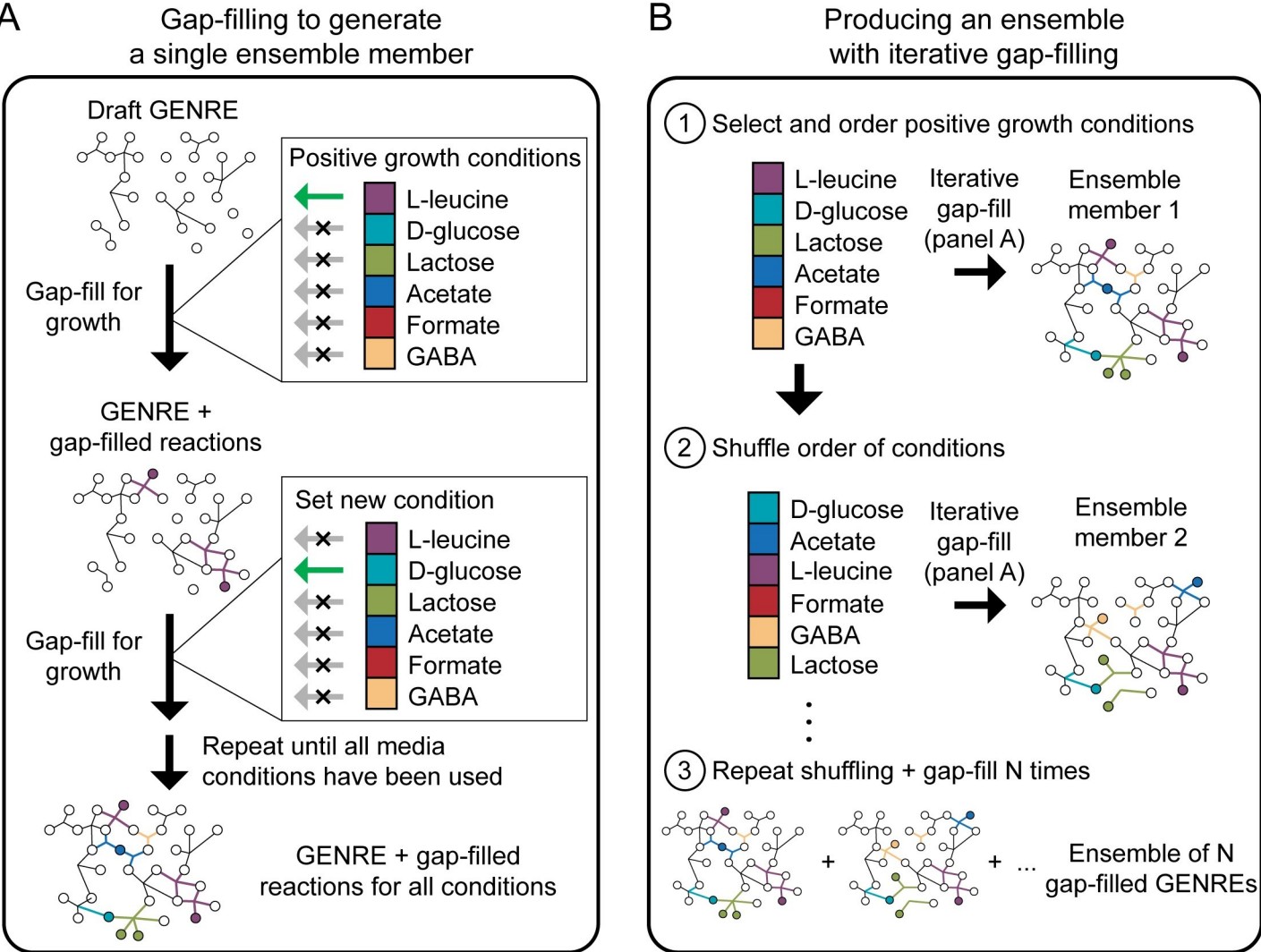

**Fig 3. Iterative gap-filling strategy used to generate ensembles implemented in Medusa. A)** Given a list of conditions in which an organism satisfied some objective (e.g., secretion of a specific metabolite, growth) and a draft GENRE for the organism, gap-filling is performed sequentially on each condition. After each gap-fill step on a single condition, reactions in the gap-fill solution are added to the GENRE before starting gap-filling on the next condition. In this schematic, all conditions are used, but Medusa also allows users to randomly subsample a fraction of conditions to generate more variation in the resulting gap-filled GENREs. Here, gap-filling is performed to enable growth in the presence of each individual metabolite (indicated by a green arrow for a single condition during each gap-fill step). **B)** Medusa iteratively performs gap-filling as shown in panel **A**, then shuffles the order of conditions to introduce variation in the gap-fill solution. After a user-defined number of cycles through the process shown in panel **A**, Medusa generates an ensemble containing all unique GENREs that resulted from gap-filling.

(i.e. the species does not have the ability to catalyze the reaction described by the feature). Based on this examination, we disable the feature (which is a reaction in this case), perform ensemble FBA again, and find that this curation step has reduced the average predicted flux through biomass **Fig 3B**. Although we use this approach to guide curation, we also envision the same process having great utility for attributing simulation uncertainty more generically, such as with simulations performed using an ensemble generated using 'omics integration methods [11].

## Generating ensembles

The simplest way to generate an ensemble in Medusa is to load a collection of models from files or a collection of COBRApy Model objects. This can be performed by specifying a batch

of files that Medusa loads iteratively to construct an ensemble or by loading all the models into memory as COBRApy Models and passing them to an ensemble constructor function. Access to both of these methods for constructing an ensemble gives users flexibility to use other software that generates multiple GENREs and to construct their own ensembles from COBRApy Models that are only ever present in memory (i.e., never written to disk). There are also performance tradeoffs between each method; constructing an ensemble from a batch of files allows the user to specify a batch size for the number of models loaded into memory during each iteration. Increasing the batch size increases memory usage but generally decreases the time required to construct an ensemble. Similarly, constructing an ensemble from a collection of COBRApy Models already loaded into memory has high memory usage but low runtime.

Ensembles can also be generated in Medusa by performing gap-filling on an individual GENRE. Medusa implements a previously-developed algorithm for gap-filling GENREs using growth phenotyping data [6,19]. The algorithm takes a GENRE with an objective function (i.e., a COBRApy Model), a universal reaction database (stored as a COBRApy Model) and a dataset of binary growth/no-growth calls on defined media conditions as input. The objective function is then set as a constraint with bounds such that any feasible flux distribution must enable activity within the bounds (e.g. at least some amount of flux through biomass production or a demand reaction). All reactions from the universal reaction database are added to the GENRE being gap-filled, and a new objective function is set to minimize the sum of fluxes through reactions in the reaction database. The problem is solved to identify reactions taking part in this minimal flux activity. To generate a single gap-filled ensemble member, the draft GENRE is iteratively gap-filled on each positive growth media condition. This process is repeated to produce the number of desired ensemble members. Variation in model structure is introduced by randomizing the order in which media conditions are used for gap-filling. We previously found that ensemble members generated with this iterative strategy are equivalent to gap-filled models generated by finding a single gap-filling solution to all growth conditions simultaneously (i.e., with a single optimization step), yet this iterative process is multiple orders of magnitude faster than the global solution [19]. See **Fig 3A and 3B** for a schematic summarizing this approach.

This gap-filling process is implemented in Medusa through a single function, and a full example of preparing a model and all data necessary for this process are provided in the Medusa documentation at https://medusa.readthedocs.io/en/latest/creating_ensemble.html. Although Medusa implements the previously published version of this approach, it also allows users to randomly subsample a fraction of conditions to generate more variation in the resulting gap-filled GENREs. We benchmarked construction of ensembles using the standard Medusa iterative gap-filling approach, a *Staphylococcus aureus* draft metabolic network reconstruction generated via ModelSEED [26], and the modelSEED universal biochemistry using a 2019 Apple Macbook Pro (2.4GHz Intel Core i5-8279U, 16GB RAM). Ensembles with 5, 25, or 100 members, gap-filled on 10 randomized positive growth media conditions, took an average of 1.8, 5.3, and 18.6 minutes to generate in 10 independent trials (**Fig 2C**). A substantial portion of this time (~60 seconds) is spent copying the universal model; time spent after this copy step scales linearly with the size of the ensemble being generated.

## Benchmarking Medusa

The structure of a Medusa Ensemble was designed to help users simplify their code when dealing with an ensemble of models, decrease memory and disk storage demands, and decrease processing time when possible. The Medusa documentation provides thorough benchmarking examples for multiprocessing using Medusa, constructing an ensemble, and memory and disk

utilization. These benchmarking notebooks demonstrate the memory savings when using a Medusa Ensemble rather than individual models (~50MB with Medusa vs. ~16GB for individual models representing an ensemble with 1000 members) and the disk space savings in the same scenario (~6MB for a Medusa Ensemble vs. 1GB for 1000 individual models). Medusa reduces memory and storage demands by over two orders of magnitude in these cases. Improvements in runtime for performing FBA in Medusa, rather than using individual models, are more modest but improve further when using more than one processor. **Fig 2B** shows the decrease in optimization runtime in Medusa as the number of processors is increased, and the diminishing returns associated with a larger number of processors. Memory, storage, and runtime benchmarking were performed using the same hardware as the iterative gapfilling benchmarking in **Fig 2C**. All benchmarking is available in narrative-style notebooks in the Medusa documentation.

## Availability and future directions

Stable releases of Medusa are available through the Python package index (PyPI, https://pypi.org) as well as github (https://github.com/opencobra/Medusa). Documentation is available through Readthedocs at https://medusa.readthedocs.io. Current development efforts are focused on parallelization and integrating Medusa with other Python-based tools in the COBRA community.

Currently, the only high-level function available to users of Medusa to share ensembles generates a serialized version of the Ensemble object using the pickle package (the standard Python library package for serializing objects). Alternatively, users can save the base_model for any Ensemble as a Systems Biology Markup Language (SBML, [27]) file using COBRApy, then choose the formatting option of their liking to save Feature and Member information for the Ensemble. In SBML, there is not currently a standardized way to represent ensembles as required in Medusa. We plan on extending the Flux Balance Constraints package, an extension to SBML intended for constraint-based models, to enable standardized sharing of Medusa ensembles. Until then, we recommend users include both an SBML file for the base_model of each Ensemble and the serialized pickle at time of publication.

## Acknowledgments

We thank Dennis McDuffie and Jean-Luc Maigrot for feedback on the usability of Medusa and members of the Papin lab for feedback on documentation.

## Author Contributions

**Conceptualization:** Gregory L. Medlock, Jason A. Papin.

**Data curation:** Gregory L. Medlock.

**Formal analysis:** Gregory L. Medlock, Thomas J. Moutinho.

**Funding acquisition:** Gregory L. Medlock, Jason A. Papin.

**Investigation:** Gregory L. Medlock.

**Methodology:** Gregory L. Medlock, Thomas J. Moutinho.

**Project administration:** Gregory L. Medlock, Jason A. Papin.

**Resources:** Gregory L. Medlock, Jason A. Papin.

**Software:** Gregory L. Medlock, Thomas J. Moutinho.

**Supervision:** Jason A. Papin.

**Validation:** Gregory L. Medlock.

**Visualization:** Gregory L. Medlock.

**Writing – original draft:** Gregory L. Medlock.

**Writing – review & editing:** Gregory L. Medlock, Thomas J. Moutinho, Jason A. Papin.

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
