## [Decision Letter · Decision Letter 0]

22 Oct 2019

Dear Jason,

Thank you very much for submitting your manuscript, 'Medusa: software to build and analyze ensembles of genome-scale metabolic network reconstructions', to PLOS Computational Biology. As with all papers submitted to the journal, yours was fully evaluated by the PLOS Computational Biology editorial team, and in this case, by independent peer reviewers. The reviewers appreciated the attention to an important topic but identified some aspects of the manuscript that should be improved.

We would therefore like to ask you to modify the manuscript according to the review recommendations before we can consider your manuscript for acceptance. Your revisions should address all the specific points made by each reviewer and we encourage you to respond to particular issues. In particular, please be sure to address the points about quantification of the performances and gains of MEDUSA.

Please note while forming your response, if your article is accepted, you may have the opportunity to make the peer review history publicly available. The record will include editor decision letters (with reviews) and your responses to reviewer comments. If eligible, we will contact you to opt in or out.raised.

- Supporting Information uploaded as separate files, titled 'Dataset', 'Figure', 'Table', 'Text', 'Protocol', 'Audio', or 'Video'.

We hope to receive your revised manuscript within the next 30 days. If you anticipate any delay in its return, we ask that you let us know the expected resubmission date by email at ploscompbiol@plos.org.

Sincerely,

Hugues Berry

Associate Editor

PLOS Computational Biology

Douglas Lauffenburger

Deputy Editor

PLOS Computational Biology

[LINK]

Reviewer's Responses to Questions

**Comments to the Authors:**

Reviewer #1: This software paper describes the Medusa tool, a software to build and analyse ensembles of genome-scale metabolic network reconstructions. The main interest of Medusa is to enable the simultaneous run and comparison of analyses over families of metabolic network, such as FBA, FVA and gene essentiality. To that goal, a family of metabolic networks is represented in CobraPy Python package using a so-called Ensemble class, which references both a CobrayPy Model (describing the base model) and Features and Members to describes the specificities of the family of metabolic networks. This smart representation allow comparing ensemble of simulations and running pipelines which alternate analyses of families of metabolic networks, then selection of sub-families, followed by novel analysis. An example of such a pipeline is a curation procedure based on the prioritizing of components which contribute the most to simulation uncertainty. Another example is the automatic generation of a family of gap-filled metabolic models according to different media conditions allowing the user to see the impact of the order of the selected media conditions over the gapfilling procedure.

As stated by the author, this representation of families of metabolic networks which relevant classes in CobraPy is a very smart solution to run multiple analyses and workflows and study the variability of metabolic networks while avoiding to store the multiple networks generated along the pipelines. This is particularly useful for automatic analyses involving the generation of multiple models.

However, the paper is lacking some elements to ensure that the Medusa package can be used by the community. They correspond to the four main limitations below that needs to be clarified.

(1) There is no information about the gains enabled by Medusa. What is the maximum number of metabolic networks that an Ensemble can describe ? What is the gain of storage space enabled by the ENSEMBLE representation with respect to a usual representation with individual models ? What are the gains of performance for FBA, FVA, and gene essentiality computations with respect to simulations with individual models ?

(2) Few details are given about the generation of ENSEMBLE. How can an ENSEMBLE be generated from a family of SBML files generated with other tools ? Does the package enable this ? In addition, the author suggest to store share ENSEMBLES by using a SBML file for the base_model and then to serialize the ENSEMBLE. What is the gain if this is the recommended solution ?

(3) Figure 4 is quite misleading. What is the advantage of generating Ensembles of gapfilled GENREs by shuffling the different growth conditions ? May the author show FBA and FVA simulations evidencing that the study of the Ensemble generates gains in the study of the considered metabolic network ?

(4) In the introduction, the authors advocate that ENSEMBLEs are useful to handle alternative solutions generated by simulations, including infinite number of feasible flux values within the range determined by a FBA solution space. I do not understand how MEDUSA is able to handle this issue ? Is this possible to generate an ENSEMBLE modeling all such GENREs with alternative feasible fluxes and then to study them with MEDUSA ? How can this be done ? If this is not the case, I would rather suggest to remove the paragraph in the introduction to avoid confusion.

Reviewer #2: The work presented here represents a useful tool for the community. However, there are some points in the manuscript that require clarification.

• In the example provided in the “Coupling ensemble modeling with machine learning,” it is not clear what data the random forest is trained on. Where it says that the “… random forest regressor is used to predict the simulated values …,” are the authors actually training the random forest on those simulations and then going in to the decision trees to determine the most predictive features? If so, this should be clarified. If not, then the information needs to be added.

• Some sense of the simulation “wall time” would be helpful for various size models/ensembles. Of course, it will also be necessary to provide information regarding hardware used in order for the wall time to have any significance.

• One assumes that Medusa is based on Python 3, and it will become obvious once an individual uses the package. However, it would still be worthwhile to mention the version of Python against which this package is built against.

Reviewer #3: In this paper, the authors describe and demonstrate Medusa, a (Python) tool for the analysis of ensemble of metabolic networks. While constraint-based modelling methods have gained immense popularity over the last two decades, they are still beset with a key problem of "degeneracy" in solutions. Furthermore, there exists uncertainty in network structure, in terms of the topology, and many gap-filling algorithms exist to fill gaps (identify missing reactions) in metabolic networks. Building on COBRAPy, Medusa can perform ensembleFBA, to address these and other important issues. Medusa also lends itself to applying machine learning tools for improving curation, and the documentation lists an example of this too.

I found the paper to be very well-written and crisp. It gives a very clear overview of the use of Medusa, and the underlying methods.

Minor Revisions:

1. It would be nice to describe an example of how parallelisation affords advantages, and also a brief discussion of how it scales. It would be advantageous to the users to be well aware of the parallel features of Medusa, which will enable them to work with larger numbers of simulations (or larger networks).

2. While Figure 3 is very good and explanative, I would find it interesting to look at a scatterplot of difference in biomass growth rates vs. similarity of the reaction networks in the ensemble. That is, for every pair of networks in the ensemble, plot a point corresponding to (d(network1, network2), v_bio1-vbio2). Is there a clear correlation? What would the outliers, if any, mean? For distance, a Jaccard distance could be used.

3. An example (both in the manuscript) and documentation, as to how ensemble performance can be studied by plotting ROC curves in Medusa could be described.

**Have all data underlying the figures and results presented in the manuscript been provided?**

Reviewer #1: Yes

Reviewer #2: Yes

Reviewer #3: Yes

PLOS authors have the option to publish the peer review history of their article (what does this mean?). If published, this will include your full peer review and any attached files.

Reviewer #1: No

Reviewer #2: No

Reviewer #3: No

---

## [Decision Letter · Decision Letter 1]

17 Mar 2020

Dear Jason,

Thank you very much for submitting your manuscript "Medusa: software to build and analyze ensembles of genome-scale metabolic network reconstructions" for consideration at PLOS Computational Biology. Based on the reviews, we are very likely to ultimately accept this manuscript for publication; however, toward that end we ask that you consider a minor revision of the manuscript to address the final recommendation made by reviewer #1 (about the citation of Biggs & Papin, PLOS Comp Bio 2017, see below).

Sincerely,

Hugues Berry

Associate Editor

PLOS Computational Biology

Douglas Lauffenburger

Deputy Editor

PLOS Computational Biology

[LINK]

Reviewer's Responses to Questions

**Comments to the Authors:**

Reviewer #1: The authors have addressed most of my comments. However, they decided to not directly address my comment on the former figure 4 (now figure 3) about the underlying motivations for the choice of the gap-filling method implemented in medusa to generate ensembl. Although I understand their position, I would suggest as a last minor revision to clearly refer to the paper Biggs & Papin, PLOS Comp Bio 2017 when they introduce Figure 3 (paragraph that starts with "Ensembles can also be generated in Medusa by performing gap-filling on an individual GENRE..." to include a summary of the detailed motivation that they provided in the response to reviewed.

Reviewer #2: The authors have addressed all of my concerns.

Reviewer #3: All my comments have been addressed satisfactorily, more or less. I am happy to recommend the publication of the manuscript.

**Have all data underlying the figures and results presented in the manuscript been provided?**

Reviewer #1: Yes

Reviewer #2: Yes

Reviewer #3: Yes

PLOS authors have the option to publish the peer review history of their article (what does this mean?). If published, this will include your full peer review and any attached files.

Reviewer #1: No

Reviewer #2: No

Reviewer #3: No
---

## [Editor Report · Decision Letter 2]

3 Apr 2020

Dear Jason,

We are pleased to inform you that your manuscript 'Medusa: software to build and analyze ensembles of genome-scale metabolic network reconstructions' has been provisionally accepted for publication in PLOS Computational Biology.

Best regards,

Hugues Berry

Associate Editor

PLOS Computational Biology

Douglas Lauffenburger

Deputy Editor

PLOS Computational Biology

---

## [Editor Report · Acceptance letter]

21 Apr 2020

PCOMPBIOL-D-19-01338R2 

Medusa: software to build and analyze ensembles of genome-scale metabolic network reconstructions

Dear Dr Papin,

I am pleased to inform you that your manuscript has been formally accepted for publication in PLOS Computational Biology. Your manuscript is now with our production department and you will be notified of the publication date in due course.

With kind regards,

Laura Mallard
